# Genetic Progress Achieved during 10 Years of Selective Breeding for Honeybee Traits of Interest to the Beekeeping Industry

**Ségolène Maucourt [1,\*], Frédéric Fortin [2], Claude Robert [3] and Pierre Giovenazzo [1]**

[1] Department of Biology, Vachon Pavillon, Université Laval, Québec, QC G1V 0A6, Canada; pierre.giovenazzo@bio.ulaval.ca

[2] Centre du Développement du Porc du Québec, 450-2590 Boulevard Laurier, Québec, QC G1V 4M6, Canada; ffortin@cdpq.ca

[3] Department of Animal Science, Institut sur la Nutrition et les Aliments Fonctionnels Pavillon, Université Laval, Québec, QC G1V 0A6, Canada; Claude.Robert@fsaa.ulaval.ca

\* Correspondence: segolene.maucourt.1@ulaval.ca; Tel.: +1-581-988-1501

**Abstract:** Genetic improvement programs have resulted in spectacular productivity gains for most animal species in recent years. The introduction of quantitative genetics and the use of statistical models have played a fundamental role in achieving these advances. For the honeybee (*Apis mellifera*), genetic improvement programs are still rare worldwide. Indeed, genetic and reproductive characteristics are more complex in honeybees than in other animal species, which presents additional challenges for access genetic selection. In recent years, advances in informatics have allowed statistical modelling of the honeybee, notably with the BLUP-animal model, and access to genetic selection for this species is possible now. The aim of this project was to present the genetic progress of several traits of interest to the Canadian beekeeping industry (hygienic behavior, honey production and spring development) achieved in our selection program since 2010. Our results show an improvement of 0.30% per year for hygienic behavior, 0.63 kg per year for honey production and 164 brood cells per year for spring development. These advances have opened a new era for our breeding program and sharing this superior genetic available to beekeepers will contribute to the sustainability and self-sufficiency of the beekeeping industry in Canada.

**Keywords:** BLUP-animal model; honeybee; selection; breeding values; breeding program; hygienic behavior; honey production; spring development





## 1. Introduction

At about the same time humans domesticated animals for food, they began a selection process, choosing individuals that exhibited specific traits. Domestication and selection are even considered indissociable because selecting animals according to their docility led to breeding and an improvement of animal husbandry [1–3]. Human societies have been exercising empirical selection on farm animals since the Neolithic period but systematically-structured selection programs did not appear until the mid-20th century, with the development of quantitative genetics and the Best Linear Unbiased Prediction (BLUP)-Animal model [4]. This statistical method is an international standard in breeding programs that has led to an improvement of up to 63% obtained in the response to selection in some animal production programs [5]. The BLUP-animal model consists of estimating the genetic component of an animal's performance (its breeding value) based on the performance of the individual, its parents, and their degree of kinship, while eliminating non-genetic effects (effects of the environment) as effectively as possible [3,4,6].

Breeding honeybees is challenging due to their haplo-diploïd reproduction, genetic architecture, sensitivity to inbreeding and social nature, which define them as super organisms [7–9]. These features explain why selection in beekeeping has not advanced as fast as

in other animal productions [10–12]. Indeed, the statistical models used in animal selection have long remained difficult to transpose to the genetic model of the honeybee [13–16]. However, recent advances in computer science and statistics offer new possibilities that can be used with quantitative genetics, such as the BLUP-animal model adapted to the honeybee's reproductive biology [17,18].

Over the past several years, our research team has been using this new technology to modernize our honeybee breeding program and adapt it to the specific conditions of Canadian beekeeping. Indeed, most Canadian honeybee farms experience short, warm summers (June–August) and long, cold winters (October–April) [19–22]. These climatic conditions shape honeybee colony management and favor the selection of hardy strains of honeybees that survive long and cold winters and develop rapidly in spring [22,23]. In addition, Canadian beekeeping has been experiencing significant colony winter mortality since 2007, at an average level of 26% per year [24,25]. These abnormal losses stem from multiple causes, such as parasitism and diseases, impoverishment of floral resources, exposure to pesticides, or stress associated with pollination services [24,26,27]. To address this problem, Canadian beekeepers import honeybees and queens from abroad, mainly from Australia, New Zealand, Chile and USA (California or Hawaii) [28,29]. However, imported honeybee strains are not adapted to northern conditions or Canadian beekeeping management practices, resulting in low winter survival rates, increased queen mortality, and reduced overall colony productivity [22,30,31]. Thus, honeybee importation compromises self-sufficiency efforts, food security and the sustainability of the country's beekeeping enterprises.

Supporting and developing honeybee breeding programs is a sustainable solution, as it allows the maintenance of local honeybee stocks while improving bee production, health status and hardiness, which are characteristics greatly prized by Canadian beekeepers [29]. In addition, the integration of the BLUP-animal methodology in honeybee breeding offers greater precision in the selection process of breeder colonies and accelerates the genetic progress of the selected traits because it is a better estimate of genetic merit, whereas phenotypes include the contributions of environmental conditions. Indeed, applying the BLUP-animal model to the honeybee generates a genetic evaluation for each colony in a breeding program and integrates the performance results of all related colonies for the trait being selected [17,18]. The "Beebreed" European program is the first to have used BLUP-animal, and results over the past years show the improvement of several traits in honeybees, such as honey production, gentleness, swarming tendency and varroa resistance [15,17,32,33].

In this study, we present genetic improvements achieved during 10 years of selective breeding for several traits of interest to Canadian beekeeping: hygienic behavior, honey production, and spring development. The study was carried out within our selection program at the University Laval-"Centre de Recherche en Science Animal de Deschambault" honeybee research center (UL-CRSAD). This program was initially based on phenotypic mass selection, which was replaced in 2016 by the BLUP-animal technology. To our knowledge, this selection program is the first in North America to integrate the BLUP-animal model to improve the genetics of the honeybee under northern conditions.

## 2. Materials and Methods

### 2.1. Biological Material

Our study was conducted in honeybee colonies at the UL-CRSAD Québec, Canada (N 46°40.27′, W 10°71.50′). The breeding program started in 2010, with 26 colonies. Eleven of these colonies' queens were from local breeders in Quebec or elsewhere in Canada with European-derived stock, and the 15 others were from Buckfast lines imported from Denmark (Buckfast Denmark [34]). From among these colonies, seven were selected for queen production and 10 for drone production. Mating these selected colonies together made it possible to produce about one hundred queens (12 per maternal lineage) in 2011. The entire resulting breeding program is therefore based on this common ancestral stock. Genetic selection for this study was conducted in this closed population, with the exception

of only three introduced lines from local queen producers: two lines in 2018 and one in 2019.

### 2.2. Traits Measured in Colonies

The traits studied were chosen based on the requirements and challenges reported by the Canadian beekeeping industry [29,35,36]. The traits measured for this study are hygienic behavior, honey production, and spring development. All measurements were recorded in a performance database. Heritability of these traits and genetic correlations were estimated in a previous study by our research team [37].

#### 2.2.1. Hygienic Behavior

To test hygienic behavior, the colony was opened, and a comb containing a solid patch of sealed worker brood at the pupal stage, with pupae having pink or purple eye color, was selected. Two PVC tubes (2″ inside diameter) were pressed down to the midrib of the comb. The number of empty cells ("misses") in each tube was counted. Liquid nitrogen was then applied, 300 mL/tube, to "freeze kill" the brood. Frames were marked and returned to the colony. After 24 h, the number of cells that remained capped or partially removed was counted. The total number of cells removed was counted, yielding a percentage of hygienic behavior [38,39]. Between 2011 and 2015, this measurement was taken twice before the first honey flow, at a two-week interval at the end of May or in June. In 2015, the hygienic behavior test was not performed on all colonies of our honeybee population. Since 2016, measurements have been taken twice after the summer honey flow during August of each year, with a two-week interval. The mean of the two tests yielded the colony's hygienic behavior rate [40].

#### 2.2.2. Honey Production

Each colony was provided with honey supers, which were placed above the brood chamber and separated by a queen excluder. The honey supers were added during the honey flow according to the productivity of each colony. Honey weight gain was obtained by weighing honey supers when added and removed from a colony, or by placing the entire colony (brood chamber and honey supers) on a platform scale (CAS-USA, East-Rutherford, NY, USA; CAS CI-2001BS) at those two times. Honey production of a colony was calculated by adding the gain in honey obtained in each honey super placed in that colony.

#### 2.2.3. Spring Development

Colony strength was evaluated by measuring the area occupied by immature worker honeybees (eggs + larvae + capped brood) in colonies in early June. This was done by measuring the width and length of the brood surface area on each side of every brood frame. The rectangular surface obtained was multiplied by 0.8 to compensate for the elliptic shape of the brood pattern. These values were added to calculate the total brood surface in each colony. A factor of 25 worker cells per 6.25 cm$^2$ (i.e., a square inch) was used to convert the area to obtain the number of immature worker honeybees [41,42].

### 2.3. Selection from Breeding Value and Selection Index
#### 2.3.1. CRSAD Breeding Program Selection Plan Launched in 2010

Our initial breeding program, begun in 2011, was based on mass selection. It consisted of selecting colonies for breeding solely on colony phenotypic performance (e.g., honey production or hygienic behavior) of each generation. Each year, between 7 and 14 colonies were selected for queen production (i.e., mothers-of-queens or queen-producing colonies) (Table 1) and 10 colonies were selected for drone production. From each queen-producing colony, 10 to 12 sister queens were reared and introduced into new colonies for the next generation.

**Table 1.** Number of colonies and number of queen-producing colonies in each generation in the UL-CRSAD breeding program. Phenotypic measurements were taken each year in all colonies.

| | Generation | Number of Colonies | Number of Queen-Producing Colonies for Breeding Next Generation |
|---|---|---|---|
| Selection based on phenotypic measurements | 2010 | 26 | 7 |
| | 2011 | 60 | 11 |
| | 2012 | 38 | 12 |
| | 2013 | 45 | 13 |
| | 2014 | 109 | 11 |
| | 2015 | 144 | 14 |
| Selection with BLUP-animal model | 2016 | 97 | 9 |
| | 2017 | 85 | 11 |
| | 2018 | 152 | 9 |
| | 2019 | 134 | 8 |

Starting in 2016, the selection of breeding colonies has been integrated the BLUP-animal technology to determine selection indexes that incorporate the breeding value of each trait measured (see Section 2.3.6). This novel index can select colonies for queen production (Table 1). From each of these colonies, we produce 15 sister queens to create at least 100 colonies per generation per year. Performance traits are measured for all these new colonies with an overlap of two years. Indeed, the generation interval within our breeding program is two years: the colonies are evaluated on their performance traits (one full beekeeping season: Summer-Fall-Winter-Spring) and the next summer, best performing colonies are selected for the next generation of queens/lines. Therefore, it is important to anticipate future winter colony losses so that the number of colonies for each queen-producing colony is sufficient for the selection process and performance improvement [43–45]. A detailed description of the selection and breeding procedure is provided in Maucourt et al. [37].

The choice of colonies for breeding males was also redesigned after the introduction of BLUP-animal in 2016. Prior to 2016, the kinship of the drone-producing colonies was not considered, and it was thus impossible to account for the male pedigree when estimating the breeding value of the different selection traits [6,46]. Since the integration of the BLUP-animal model in our selection program, related drone-producing colonies are selected which means all the queens are sisters, thus all from the same mother, who is also the grandmother of these cousin drones (Figure 1). These related males are considered as dummy fathers that can be integrated into the statistical model [17,47].

To ensure proper mating of virgin queens, a minimum of six drone-producing colonies with sister queens are necessary to form the dummy father. According to the literature, each colony produces enough mature males to fertilize 35 queens per summer month, so at least six drone-producing colonies are needed to fertilize the 200 virgin queens per summer month [48].

All genealogical information pertaining to the colonies in our breeding program is recorded in a pedigree database which is used by our BLUP-animal statistical model. The genealogical tracking of the different generations of colonies operated in our pedigree database is only based on the queens. For each selected queen, we have identified her mother, sisters, and daughters [10]. Since 2016, dummy fathers involved in reproduction are also inserted in the pedigree database using their grandmother's pedigree information (see Table 1) [17]. All queens associated with our breeding program are identified by a unique identification number [49]. This identification number is used in the pedigree database and the performance database [17,47,49]. Each queen's unique identifier includes three elements: year of birth of the queen (last two digits), breeder code (one letter) and queen number within the studbook of the breeder (three digits).

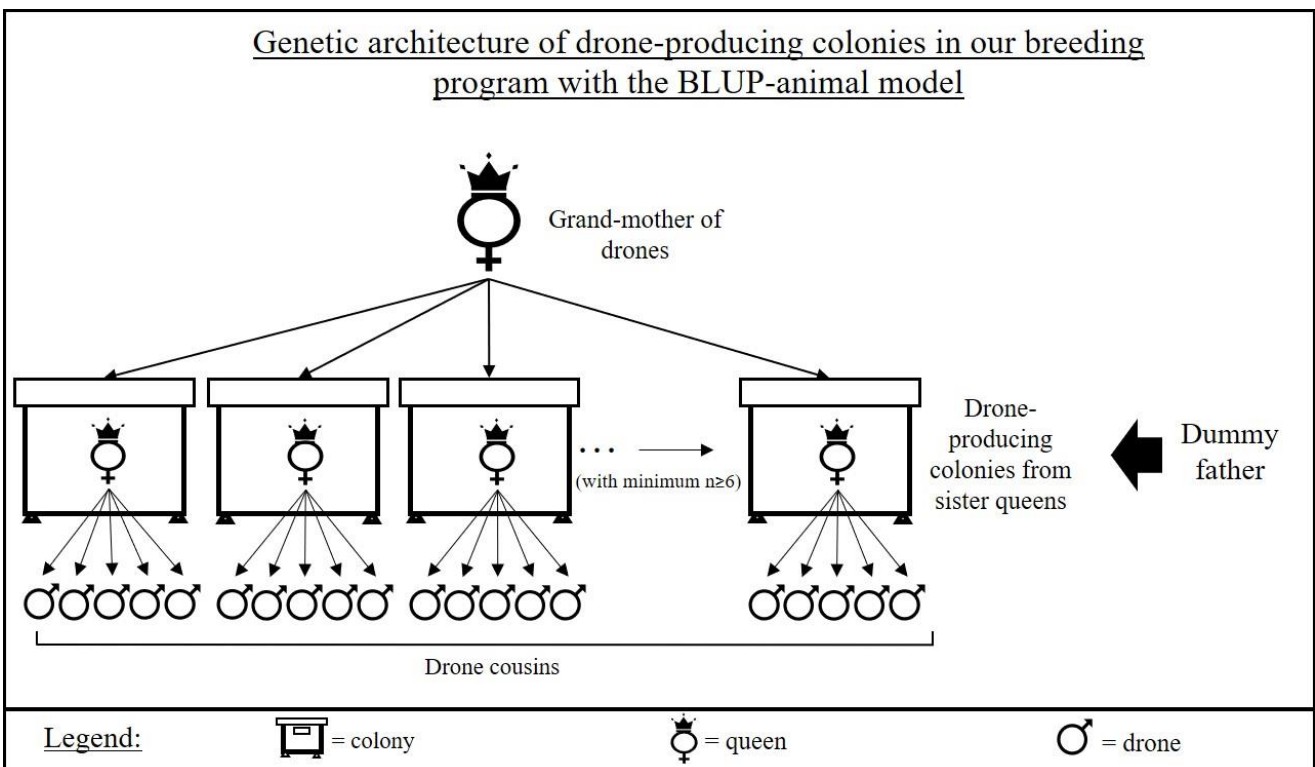

**Figure 1.** Model representing the dummy father used in our selection program with the BLUP-animal model. The dummy father is composed of a group of drone-producing colonies from sister queens. These queens from the same mother make it possible to obtain related drones (cousin drones) for reproduction.

#### 2.3.2. Rearing and Mating Process

Each year, young queens are produced from the selected mother lines (12 per generation until 2016, and 15 per generation thereafter) using the Doolittle queen rearing method [50]. Each royal cell is introduced in a mating nuc (dimensions: 12 1/2″ × 7 7/8 × 8 5/8″, Propolis-etc..., Saint-Pie, QC, Canada; MN-1200) comprised of two brood frames with the young adherent bees, an empty frame with drawn cells, and a frame with honey and pollen). Mating nucs are placed in an apiary located 1.2 km from drone-producing colonies and 1.6 km from the drone congregation area [51]. Furthermore, a drone frame (Propolis-etc..., Saint-Pie, QC, Canada; PL-1900) is placed in the center of the brood chamber in each drone-producing colony to increase the populations of selected drones. Our drone control methodology provides massive flooding of selected drones and ensures that 83 to 93% of mating is performed by the selected males [52–55].

#### 2.3.3. Guaranteed Selected Genetics

Several precautions are taken with regard to queens and colonies from the UL-CRSAD breeding program to ensure the authenticity of each queen's pedigree and also the validity of her performance data in association with her colony [56,57]. All queens in the UL-CRSAD breeding program are identified by marking the back of their thorax with a queen marking pen (Propolis-etc..., Saint-Pie, QC, Canada; MP-1103 to MP-1104) and clipping half of one of their two anterior wings to prevent them from swarming. During the swarming period, colonies are inspected every 15 days to destroy queen cells. If a queen swarms or dies before or during the performance measurement period, the colony is excluded from the breeding program and performance is not measured after this event.

#### 2.3.4. Colony Management

Each year, the queens produced by breeder colonies are introduced in double nuclei Langstroth four frames (Propolis-etc..., Saint-Pie, QC, Canada; NU-2002) comprising two

full brood frames with their young adherent honeybees, an empty frame with drawn cells and, a food frame with honey and pollen. In September, each nucleus colony is treated against *Varroa destructor* (Thymovar®, Propolis-etc..., Saint-Pie, QC, Canada; TH-1110;and Apivar®, Propolis-etc..., Saint-Pie, QC, Canada; AP-2000), fed 10 L of sucrose-water solution (2:1) using a double nuclei feeder with floaters (Propolis-etc..., Saint-Pie, QC, Canada; FE-1700) and overwintered (starting mid-November) in an environmentally controlled room (4 ± 1 °C and 40–50% RH). The following year, in early May, the queens, bees and frames of each nucleus colony are transferred into Langstroth 10-frame hives. Colonies are then equally distributed between four (from 2010 to 2016) or five (since 2016) different apiaries according to the number of surviving colonies. Each year, colonies are distributed so that sister-queens of all the lines are present in all apiaries used [49,57]. These apiaries are situated within a radius of 40 km of our research center (CRSAD) and at least 3 km apart from each other in a similar agricultural environment with the same potential honey production (Figure 2). These colonies are managed for honey production and their performance is evaluated during summer. In September, colonies are again treated against *Varroa destructor*, fed with 20 L of sucrose-water solution (2:1) and overwintered (starting mid-November) in an environmentally controlled room (4 ± 1 °C and 40–50% RH). The following spring, the surviving colonies are ranked according to performance and breeder colonies are selected (queen-producing colonies and drone-producing colonies).

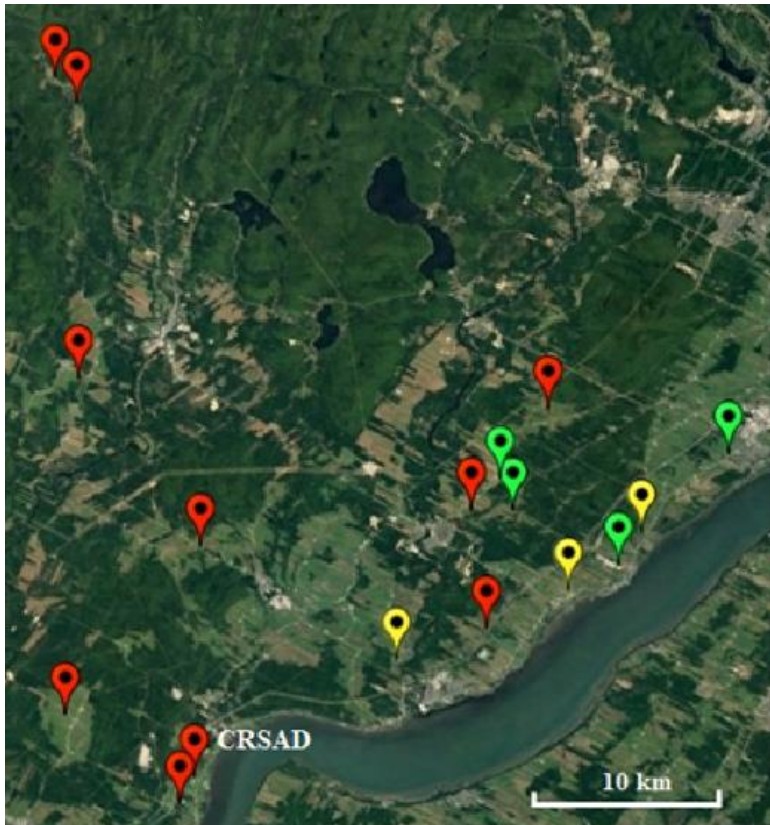

**Figure 2.** Map of the location of the different apiaries used for the colonies of the CRSAD selection pro-gram. From 2010 to 2016, these colonies were randomly distributed on four of these apiaries, and, since 2016, on five. Each year, the location of some of the apiaries used for the selection program changed. The tags in red correspond to the location of apiaries used between 2010 and 2016, in yellow to the location of apiaries used between 2010 and 2019 and in green to the location of apiaries used between 2016 and 2019 (Source: screenshot of Google TM Earth).

### 2.3.5. Statistical Analysis

Statistical tests were conducted using ASReml-R software (ver. 4.1.0.143, VSNi Inc., Hemel Hempstead, England, UK) using the database containing the performance associated to colonies in our breeding program and the inverse of the pedigree relationship matrix obtained from the pedigree database. All dependent variables were tested for normality using the Skewness and Kurtosis test, and no Box-Cox power transformations were required to meet the assumptions of model normality. Breeding values of colonies were estimated with the BLUP-animal model adapted to honeybees. This statistical model, used for estimated the breeding value of each colony for each trait, is the following:

$$y = Xb + Z_1u_1 + Z_2u_2 + e \tag{1}$$

where y = performance trait (i.e., honey production, spring development, or hygienic behavior); X = incidence matrix relating the fixed effects (corresponding to colony environment: apiary and year); b = vector of fixed effects (apiary and year); $Z_1$ = incidence matrix relating observations to the corresponding worker effects; $u_1$ = vector of random worker effects; $Z_2$ = incidence matrix relating observations to the corresponding queen effects $u_2$ = vector of random queen effects and e = vector of random residual effects associated with the measurements.

Before 2016, drone-producing colonies were not related, and thus the paternal side in the pedigree database was absent from the statistical model [18]. Since 2016, drone-producing colonies are related and integrated as a dummy father at each generation in the model (Figure 1). The maternal affiliation of the dummy father in the pedigree database corresponds to the mother of the group of sister drone-producing colonies [17,47]. This adjustment in the pedigree of the colonies offers pedigree information of the fathers and the mothers, thus maximizing the selection response.

### 2.3.6. Choice of Breeders

For each generation, a breeding value is calculated for each colony and the various performance traits (hygienic behavior, spring development and honey production). Thus, to select several traits simultaneously, a selection index is applied [6,44,45,58]. A selection index allows assigning a value for each trait according to its economic importance, heritability and genetic correlations with other traits [18,59]. In our breeding program, we used the following selection index:

$$I_y = BV_{hb_y} \times 0.5 + BV_{hp_y} \times 0.3 + BV_{sd_y} \times 0.2 \tag{2}$$

where, $I_y$ = selection index of colony $y$; $BV_{hb_y}$ = standard deviations of the breeding value of hygienic behavior of colony $y$; $BV_{hp_y}$ = standard deviations of the breeding value of honey production of colony $y$; and $BV_{sd_y}$ = standard deviations of the breeding value of spring development of colony $y$. In our breeding program, we have decided to give the most importance to resistance to pests and diseases, because it is an economically important trait for the beekeeping industry [18,59,60]. Therefore, the weighting coefficient associated with hygienic behavior is 0.5, which is 50% of our total selection index. We then assign a weighting coefficient of 0.3 to honey production and 0.2 for spring development. These are all important traits for Canadian beekeeping [29,61,62].

Each year, the first 10 colonies of this ranking are selected as breeders (queen-producing colonies and drone-producing colonies). The drone-producing colonies are selected according to the selection index of their mother (or grandmother of the drones, see Figure 1) in order to choose the group of sister colonies with the best genetic potential [17,47].

## 3. Results

Over 2000 data inputs have been recorded in our UL-CRSAD breeding program since 2010 (Figure 3). Since 2016, our database has increased to 350 data inputs per year for these three selection traits.

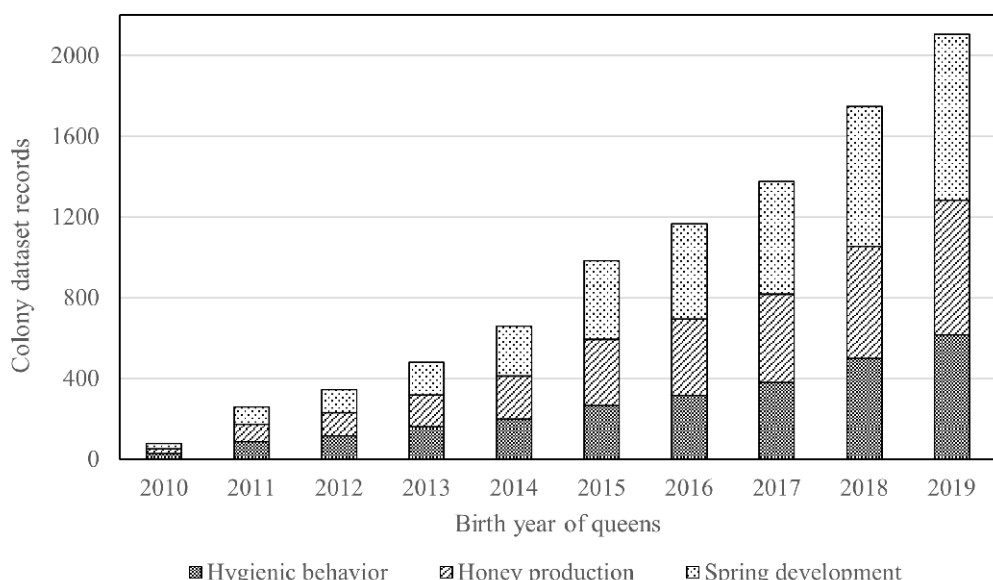

**Figure 3.** Development of the databank for the performance of hygienic behavior, honey production and spring development traits used to calculate breeding values in the UL-CRSAD honeybee breeding program.

Selection conducted in the breeding program for the three performance traits since 2010 shows significant genetic improvement over the 10-year period (Figure 4). For hygienic behavior, there is an average genetic progress of 0.3% per year, for honey production an average genetic progress of 0.6 kg of honey per year and for spring development an average genetic progress of 164 brood cells per year. When we compare phenotypic selection (Figure 4, dark grey triangles) with BLUP-animal methodology (Figure 4, light grey diamonds), the genetic improvement rate has increased fourfold for spring development, nearly twofold for honey production, and has decreased slightly for hygienic behavior.

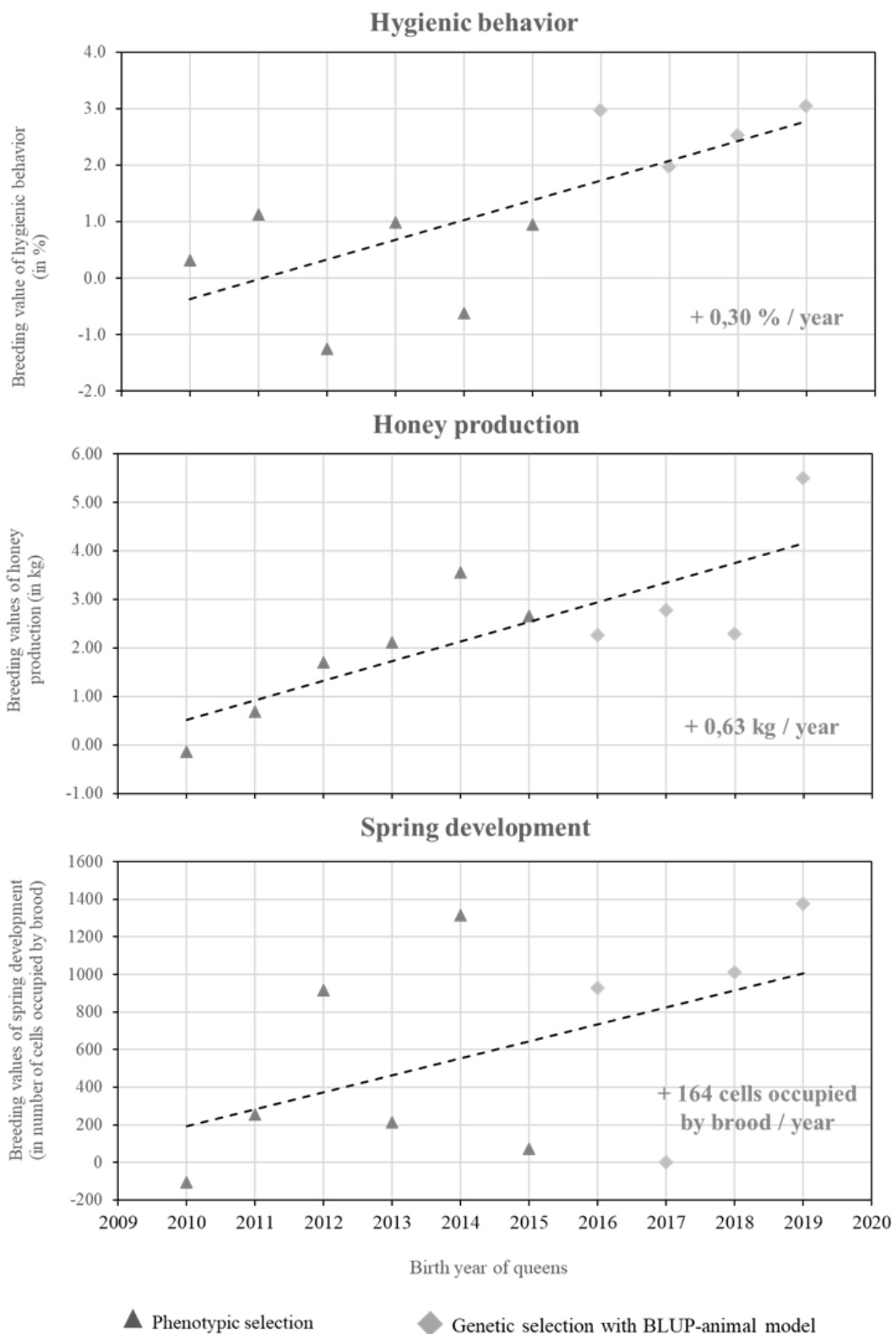

**Figure 4.** Genetic improvement for three traits: hygienic behavior, honey production and spring development in the UL-CRSAD honeybee breeding program since 2010. Each point represents average breeding value of all colonies by year. The dark grey triangles represent the average breeding value per year before introduction of the BLUP-animal methodology and the light grey diamonds represent the average breeding value per year following its introduction. The average annual improvement for each trait (regression coefficients) is shown in each graph at the bottom right.

## 4. Discussion

The aim of this project was to present the data on 10 years of genetic improvement of selective breeding for several traits of interest for Canadian beekeeping, hygienic behavior, honey production, and spring development. To achieve this, we estimated the breeding values of all the colonies in the selection program since 2010 using the BLUP-animal technology. Our work was carried out within the context of our selection program at the UL-CRSAD honeybee research center.

Our measurements showed significant genetic improvement of the performance traits since their selection began in 2010. Indeed, we found that honey production increased 0.6 kg per year, which is quite similar to the genetic progress achieved in Europe with a similar methodology. There, the "Beebreed" program measured genetic progress ranging from 0.5 to 0.7 kg of honey per year [15,63]. Spring development and hygienic behavior also showed consequent genetic progress with respectively an increase of 164 brood cells and of 0.3% in hygienic behavior per year. Unfortunately, we have not found other results in the current literature with which to compare the results on these traits. However, even if the methods are not identical, it is interesting to see that the hygienic behavior trait measured by the pin-test method in Europe also shows a similar genetic progress, with a gain of 0.73% per year [33]. Finally, the genetic parameters of the traits selected in our selection program were evaluated in a previous study, in which we showed that the zootechnical performance of these traits could be improved through a selection program [37], which also supports the accuracy of our results.

During the last four years of our selection process, a similar trend is observed for two of the traits, with a fourfold increase in genetic progress for spring development and a twofold increase for honey production. This trend towards accelerated genetic progress can be explained by the fact that the selection program began with a mass selection strategy based on the individual performance of the colony, which resulted in modest breeding progress. The subsequent introduction of the BLUP-animal methodology allowed the selection of colonies based on estimates of breeding values that express their hereditary potential for the different selection traits. Selecting colonies based on breeding values and tight control of mating (i.e., massive flooding of selected drones cousins) provides a strong foundation for more efficient selection and therefore more important genetic progress [44,45,55]. This trend is not observed for the hygienic behavior; however, the average breeding value for the last four years is consistently higher than the average breeding value prior to the introduction of the BLUP-animal methodology. We suspect that an acceleration of genetic progress for this trait will continue to be measured in the generations to follow.

Selection using breeding values, as performed in our breeding program, is a recent practice, and thus the interpretation of the genetic progress obtained since the introduction of the BLUP-animal methodology remains fragile, as this progress has been observed over the last four years only [64,65]. However, selection based on breeding values estimated by this statistical methodology maximizes the response to selection [5] and is responsible for significant genetic progress in honeybees after several years [15,63].

In our selection program, queen-producing colonies and drone-producing colonies were selected according to an index that combines the genetic values of the selected traits. The use of a selection index allows multiple traits to be selected simultaneously in order to achieve an overall improvement in the selected colonies [66]. This multi-character selection reduces selection intensity [49], which also explains why the genetic progress for the three selected traits has remained moderate until now. Indeed, within our index we have given more importance to the hygienic behavior trait (50%) whose heritability index is considered low (i.e., $h^2 = 0.18 \pm 0.13$), as well as 30% importance to the honey production trait and 20% importance has been given to the spring development trait whose heritability indexes are considered medium (with respectively $h^2 = 0.20 \pm 0.13$ and $h^2 = 0.30 \pm 0.14$). In addition, the two traits for which we gave the least importance are also genetically correlated (r = 0.50) which means that they are genetically linked and that

they will progress genetically together [37]. However, simultaneous selection of multiple performance traits, as practiced in this study, is more aligned with the actual expectations of beekeepers than single-trait selection [29,67].

In the future, we plan to continue to evaluate performance traits using measurements rather than a classification system (i.e., traits scored on a previously established scale) to enrich the records within the databases of our breeding program. Emphasizing the quality and accuracy of phenotypic measurements will result in a more robust estimation of breeding values and therefore a selection of better-quality breeder colonies, which in turn will have a positive impact on genetic progress [49,68,69]. In addition, we will continue to introduce one or two honeybee lines from other queen producers per year to avoid an increase in the inbreeding rate within our selection program with a closed population. Furthermore, with the integration of BLUP-animal model, we have redesigned the way we select colony breeders. Indeed, before 2016, the drone-producing colonies were not related to each other, so it was impossible to insert them in the pedigree dataset of the statistical model. Without accounting for paternity in genetic relationships in a closed population, inbreeding may be underestimated over the long-term [70] or breeding value estimates may be underestimated [71]. To overcome this limitation, the selection of drone-producing colonies is done by grouping related colonies. This group of sister colonies forms a dummy father that is inserted into the pedigree to estimate the breeding values of traits [17,18]. Because of our generation interval of two years, we do not yet see the beneficial impact of complete pedigree information on our selection within the genetic progress of the selected traits, but this should become observable as of 2020.

## 5. Conclusions

The aim of this study was to present the genetic progress of several breeding traits achieved by the honeybee breeding program at CRSAD, established in 2010. The selected traits were chosen because of their importance for beekeeping in a northern context. Indeed, the climate conditions in Canada motivate beekeepers to choose hardy honeybees (rapid spring development), with high productivity (high honey production) and resistance to pathogens (high hygienic behavior). In conclusion, we have shown in this study that the traits selected thus far show significant genetic progress and that the new statistical methodology BLUP-animal is an effective tool to the selection. In the future, we will continue to select for these traits, integrate new traits in our selection index (winter consumption, resistance to the *Varroa destructor* parasite, propolis or pollen production, etc.), to develop multi-trait genetic evaluations, and offer the genetic results of our selection program to Canadian beekeepers (Figure 5). These actions will help ensure the sustainability and productivity of the beekeeping industry.

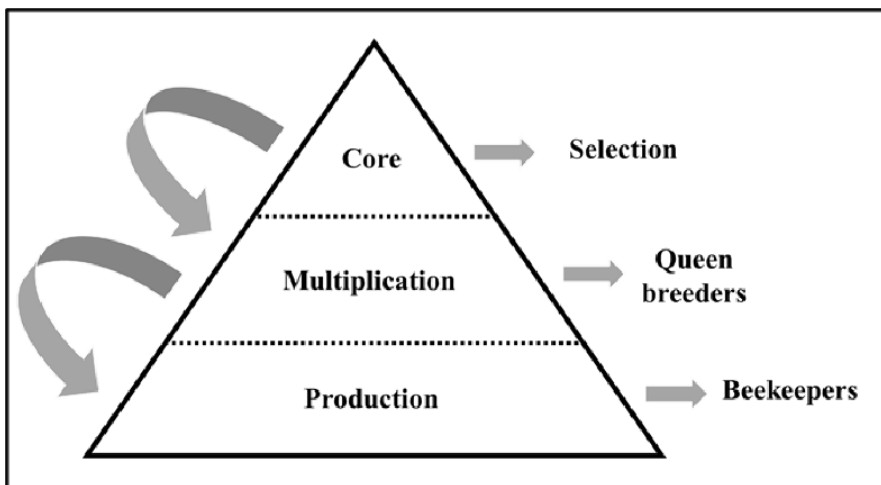

**Figure 5.** Selection pyramid and distribution patterns of genetic progress.

**Author Contributions:** Conceptualization, S.M., F.F., C.R. and P.G.; Methodology, S.M. and P.G.; Software, S.M. and F.F.; Validation, S.M., F.F., C.R. and P.G.; Formal Analysis, S.M., F.F., C.R. and P.G.; Investigation, S.M.; Resources, F.F. and C.R.; Data Curation, S.M.; Writing—Original Draft Preparation, S.M.; Writing—Review and Editing, S.M., F.F., C.R. and P.G.; Visualization, S.M.; Supervision, C.R. and P.G.; Project Administration, P.G.; Funding Acquisition, P.G. All authors have read and agreed to the published version of the manuscript.

**Funding:** This research was funded by Genome Canada (BeeOMICs, #8103), financial support was also received from the Natural Sciences and Engineering Research Council (#2019-05843) and the Centre de Recherche en Science Animale de Deschambault (#1616AP287).

**Institutional Review Board Statement:** Not applicable.

**Informed Consent Statement:** Not applicable.

**Data Availability Statement:** Data set is available upon request to the corresponding author.

**Acknowledgments:** The authors would like to thank the entire beekeeping staff of the Deschambault Research Center for Animal Sciences and the students in Pierre Giovenazzo's laboratory.

**Conflicts of Interest:** The authors declare no conflict of interest. The funders had no role in the design of the study; in the collection, analyses, or interpretation of data; in the writing of the manuscript, or in the decision to publish the results.

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
