# Peer review of "Genetic Progress Achieved during 10 Years of Selective Breeding for Honeybee Traits of Interest to the Beekeeping Industry"

_agriculture, doi:10.3390/agriculture11060535_

Round 1

Reviewer 1 Report

Comments and Suggestions for Authors

General comments

This report is a valuable contribution to the field of BLUP-supported breeding programs as it shows the potential of a regional breeding program for honeybees. The genetic yield is remarkable and will motivate breeding groups to engage in systematic breeding activities.

The manuscript is very well written.

Specific comments

L 51 Citing the more recent https://www.mdpi.com/2075-4450/11/7/404 is more appropriate.

L 61 This high number of winter losses leads to the question, why not colony survival (as a binary trait) or even overwintering ratio as a trait, because it seems to be an important issue in Canada. Give reasons why it was not considered necessary / practical in Discussion.

L 103-105 Sentence is confusing as introducing new lines is a contradiction to "closed population". Replacing "specifically using" with "with the exception" may help to clarify.

L 163 "achevae" I don't know this word

L 175 Is only a single mating facility set up? Please clarify.

If so, please discuss if this might induce problems of inbreeding in the future, and how that might be countered.

Figure 4. I feel, the high variance of Spring development BV is worth discussing.

L 323 At this point the impact of controlled mating should be mentioned which (in my viewpoint) is at least as important as using breeding values (https://gsejournal.biomedcentral.com/articles/10.1186/s12711-019-0518-y)

It should also be discussed if the open mating used in the initial years may also have provided a partial mating control (depending on other apiaries in the area) by the impact of the breeding colonies.

Author Response

Thank you for your comments and reviews of our manuscript. We have modified our manuscript with the guidance of your specific comments (below). 

Point 1: L 51 Citing the more recent https://www.mdpi.com/2075-4450/11/7/404 is more appropriate.

Response 1: We agree. This reference is added line 51.

Point 2: L 61 This high number of winter losses leads to the question, why not colony survival (as a binary trait) or even overwintering ratio as a trait, because it seems to be an important issue in Canada. Give reasons why it was not considered necessary / practical in Discussion.

Response 2: In our opinion, selection for the colony survival trait is done indirectly through the selection program without the need to use genetic data because colonies lost during winter are not reproduced the following year. Furthermore, the colony survival trait is directly dependent on the traits selected in this study: hardiness, honey production, and disease/parasite resistance.

Point 3: L 103-105 Sentence is confusing as introducing new lines is a contradiction to "closed population". Replacing "specifically using" with "with the exception" may help to clarify.

Response 3: We agree, this is confusing. We have replacing this information to the lines 103-105.

Point 4: L 163 "achevae" I don't know this word

Response 4: We agree, this is a typing error. We have modified the error to the line 163.

Point 5: L 175 Is only a single mating facility set up? Please clarify.

Response 5: All information concerning the selection and breeding procedures are referred to a previous article on our selection program. However, we have added some clarifications see line 166-167.

Point 6: If so, please discuss if this might induce problems of inbreeding in the future, and how that might be countered.

Response 6: We agree, information about inbreeding management has been added to the lines 360-362.

Point 7: Figure 4. I feel, the high variance of Spring development BV is worth discussing.

Response 7: The variance of the spring development trait in our honey bee population has been discussed thoroughly in our previous study on the heritability to the lines 315-318.

Point 8: L 323 At this point the impact of controlled mating should be mentioned which (in my viewpoint) is at least as important as using breeding values (https://gsejournal.biomedcentral.com/articles/10.1186/s12711-019-0518-y). It should also be discussed if the open mating used in the initial years may also have provided a partial mating control (depending on other apiaries in the area) by the impact of the breeding colonies.

Response 8: We agree, information and reference has been added to the lines 326-329.

Reviewer 2 Report

Manuscript at hand offers for the first results of BLUP analysis from North America and it is additional evidence on how breeding and selection in honey bees work when proper tools are applied. It is written in very nice style and language and provides clear and sound message which is supported by scientific evidence.

If I have to give a criticism, I would point out a structure of the paper: the material and methods part with 5 pages takes most of the manuscript. Some new and very important references are missing but they are suggested in the Minor changes and Suggestions part.

Minor changes and suggestions

Line 68-69 – I don't see a point of reference 29 in this statement.

Line 80-83 – I would highly suggest here to cite paper of Hoppe et al. Substantial Genetic Progress in the International Apis mellifera carnica Population Since the Implementation of Genetic Evaluation from 2020.

L120-124 – Can you give a short explanation (maybe in discussion part) why did you change the timing (month) of hygienice behaviour testing.

L176 – replace „good fertilization“ with „proper mating“.

L176 – „a minimum of at least“ doesn't sound good. I suggest to use only „minimum“ or „at least“

L211 – replace „royal cells“ with „queen cells“.

L311-312 – Here are other results in Hoppe et al. (Substantial Genetic Progress in the International Apis mellifera carnica Population Since the Implementation of Genetic Evaluation) state: The yearly genetic improvement in absolute terms (independent from the normalization basis), obtained from a linear regression on queens from 2009–2018, is 336 g for honey yield, 0.017 mark points for gentleness and for calmness, 0.018 mark points for swarming, 0.73 percent points for hygienic behavior and a reduction of 0.018 mites per 10g bee sample for VID.

L325-328 – I can see your point here. Still, in BeeBreed they are measuring hygienic behaviour with „pin-test“ method. One of the main points is that in 1995 they were checking hygienic removal after 24h and nowdays in 6 hours interval (between killing and checking the brood). Is it possible that 24 hour interval that you are using for the whole time could have an effect here? What is the average amount of cleaned cells (of all colonies) in the begining of selection and in the last year? This is just a suggestion for thinking and discussion.

Author Response

Thank you for your comments and reviews of our manuscript. We have modified our manuscript with the guidance of your specific comments (below).

Point1: Line 68-69 – I don't see a point of reference 29 in this statement.

Response 1: We agree, this is a typing error. We have modified the error to the line 68-69.

Point 2: Line 80-83 – I would highly suggest here to cite paper of Hoppe et al. Substantial Genetic Progress in the International Apis mellifera carnica Population Since the Implementation of Genetic Evaluation from 2020.

Response 2: We agree, reference has been added to the line 80-83.

Point 3: L120-124 – Can you give a short explanation (maybe in discussion part) why did you change the timing (month) of hygienice behaviour testing.

Response 3: The change in planning for hygienic behavior tests in our breeding program is only due to a colony management issue. The tests in our breeding program are always performed during the low honey flow period as suggested by Spivak & Downey, 1998 and Spivak & Reuter, 1998. In Quebec, we have two honeyflow lows, one in the spring and one in early August, and these are the periods in which we conducted our tests for this trait. It was a global change in colony management that was carried out between 2015 and 2016, i.e. all colonies were tested at the same time, either in June or August depending on the year.

Point 4: L176 – replace „good fertilization“ with „proper mating“.

Response 4: We agree, words has been changed to the line 176.

Point 5: L176 – „a minimum of at least“ doesn't sound good. I suggest to use only „minimum“ or „at least“

Response 5: We agree, “at least” has been removed to the line 176.

Point 6: L211 – replace „royal cells“ with „queen cells“.

Response 6: We agree, “royal cells” has been replaced by “queen cells” to the line 211.

Point 7: L311-312 – Here are other results in Hoppe et al. (Substantial Genetic Progress in the International Apis mellifera carnica Population Since the Implementation of Genetic Evaluation) state: The yearly genetic improvement in absolute terms (independent from the normalization basis), obtained from a linear regression on queens from 2009–2018, is 336 g for honey yield, 0.017 mark points for gentleness and for calmness, 0.018 mark points for swarming, 0.73 percent points for hygienic behavior and a reduction of 0.018 mites per 10g bee sample for VID.

Response 7: We have added these results for hygienic behavior trait in the manuscript to the lines 313-318.

Point 8: L325-328 – I can see your point here. Still, in BeeBreed they are measuring hygienic behaviour with „pin-test“ method. One of the main points is that in 1995 they were checking hygienic removal after 24h and nowdays in 6 hours interval (between killing and checking the brood). Is it possible that 24 hour interval that you are using for the whole time could have an effect here? What is the average amount of cleaned cells (of all colonies) in the begining of selection and in the last year? This is just a suggestion for thinking and discussion.

Response 8: This is a very good reflection, and indeed, it would be interesting in the future to test the effect of time (between culling and brood check) for the measurement of the hygienic behaviour trait and perhaps this "time" effect could be considered in our future selection plan. Although, a parallel study in our laboratory shows that there is no correlation between the pin-test and the freeze-brood test (these results will be published this year, contact [email protected] for more information on the subject) so we cannot really predict that the effect of time plays a role in our results as it does for the results obtained in Europe with the pin-test.